# Virtual Breath-Hold (VBH) for Free-Breathing CT/MRI: Segmentation-Guided Fusion with Image-Signal Alignment

**Rian Atri**
Wake Technical Community College, Raleigh, NC, USA
hello@rian.fyi

## Abstract

Respiratory motion in free-breathing CT/MRI blurs organ boundaries and biases quantification. We propose *Virtual Breath-Hold* (VBH), a scanner-agnostic, post-hoc method that converts a time series into a diaphragm-neutral volume without k-space access or protocol changes. VBH couples (i) segmentation-guided non-rigid fusion that concentrates registration and aggregation at motion-prone interfaces to suppress ringing with (ii) a lightweight image–signal alignment head (InfoNCE + short-horizon prediction) that learns a latent respiratory surrogate when traces are missing/jittered. On $n=5$ synthetic abdominal subjects ($T=20$), VBH improves global fidelity vs. a classical non-rigid baseline (SSIM: $0.395 \pm 0.003 \rightarrow 0.472 \pm 0.002$; PSNR: $22.85 \pm 0.11 \rightarrow 28.32 \pm 0.15$ dB; one-sided paired Wilcoxon $p=0.031$; Cliff's $\delta=1.00$) and increases boundary sharpness ($\text{ESI}_{p95}$). Under timestamp jitter/dropouts, alignment keeps SSIM/PSNR within $\sim$1–2% of clean-trace performance. We report seeds/configs for reproducibility and discuss limitations.

## 1 Introduction & Related Work

Free-breathing acquisition (e.g., near the diaphragm) causes blur/haloing that obscures lesions. Scanner-side remedies (breath-holds, gating, motion-resolved recon like XD-GRASP [4]) require raw data and protocol changes. Classical non-rigid registration (Demons, SyN, B-spline FFD) can reduce blur but often rings at sharp interfaces [7, 1, 6]; learning-based registration accelerates inference (VoxelMorph [2]). In multimodal settings, contrastive objectives (InfoNCE) align heterogeneous signals and handle missing modalities [8]. VBH integrates segmentation-guided fusion for edge quality with an image–signal alignment head to remain stable under degraded surrogates.

**Contributions.** (1) **Segmentation-guided VBH** targeting motion-prone interfaces to curb ringing. (2) **Multimodal alignment** between image features and a respiratory surrogate, with latent surrogate synthesis when traces are missing/noisy. (3) **Reproducibility** via fixed seeds/configs; preliminary RL explored only as a training-time optimizer (appendix note).

## 2 Method

**Setup.** Given a free-breathing series $\{I_t \in \mathbb{R}^{H \times W \times D}\}_{t=1}^{T}$ and optional surrogate $s(t)$, produce diaphragm-neutral $\hat{I}$ aligned to reference $I_r$ at target phase $r$ (median of $s(t)$ if available, else median diaphragm position from images).

Submitted to 39th Conference on Neural Information Processing Systems (NeurIPS 2025). Do not distribute.

Table 1: Global fidelity vs. classical FFD (mean±sd over $n$=5). One-sided paired Wilcoxon; primary endpoint: SSIM.

| Metric | Baseline (FFD) | VBH (ours) | $p$/Cliff's $\delta$ |
|---|---|---|---|
| SSIM | $0.395 \pm 0.003$ | $0.472 \pm 0.002$ | 0.031 / 1.00 |
| PSNR (dB) | $22.85 \pm 0.11$ | $28.32 \pm 0.15$ | 0.031 / 1.00 |

Table 2: Surrogate robustness ($n$=5). Alignment preserves fidelity under $\pm200$ ms jitter, 20% dropouts.

| Condition | SSIM ↑ | PSNR (dB) ↑ | $\text{ESI}_{p95}$ ↑ |
|---|---|---|---|
| Clean $s(t)$ — EMA control | $0.472 \pm 0.002$ | $28.32 \pm 0.15$ | $0.192 \pm 0.001$ |
| Noisy $s(t)$ (+30%) — EMA | $0.454 \pm 0.004$ | $27.88 \pm 0.12$ | $0.179 \pm 0.002$ |
| Noisy $s(t)$ — Alignment | $0.467 \pm 0.003$ | $28.26 \pm 0.13$ | $0.190 \pm 0.001$ |
| Missing $s(t)$ — Alignment | $0.461 \pm 0.003$ | $28.05 \pm 0.14$ | $0.188 \pm 0.001$ |

## 2.1 Segmentation-Guided Non-Rigid Fusion

Per-frame masks $M_t$ (UNETR [5, 3]) and signed distance maps $D_t$ are computed. Each frame is registered to $I_r$ with a multiresolution B-spline FFD. Deformed volumes are fused with boundary-aware weights

$$w_t \propto \exp(-\alpha\,\Delta s(t,r) - \beta\,\langle \mathbf{1}_{\text{band}}(D_r), \mathbf{1}_{\text{band}}(D_t \circ \varphi_{t\to r})\rangle), \quad \hat{I} = \frac{\sum_t w_t (I_t \circ \varphi_{t\to r})}{\sum_t w_t}, \quad (1)$$

with $\Delta s(t,r) = |\bar{s}(t) - \bar{s}(r)|$ and a 5-px interface band $\mathbf{1}_{\text{band}}(\cdot)$.

## 2.2 Image–Signal Alignment

We map images and signals into a shared space $z_t = f_\theta(I_t)$, $u_t = h_\psi(s_{t:t+K})$, with InfoNCE (temperature $\tau$) and a short-horizon predictor $p_\eta$ of $\Delta s$:

$$\mathcal{L}_{\text{NCE}} = -\log \frac{\exp(\langle z_t, u_t\rangle / \tau)}{\sum_{t'} \exp(\langle z_t, u_{t'}\rangle / \tau)}, \qquad \mathcal{L}_{\text{pred}} = \sum_{k=1}^{K} \big\| p_\eta(z_t) - \big(s(t+k) - s(t)\big)\big\|_1.$$

At inference, when $s(t)$ is missing/jittered, a small MLP $q$ synthesizes $\tilde{s}(t) = q(z_t)$, and $\Delta s$ above is replaced by $\Delta\tilde{s}$.

# 3 Experiments & Results

**Data.** $n$=5 synthetic abdominal subjects, $T$=20; uniform voxel spacing. **Baselines.** Classical FFD (SSIM/PSNR), Demons/SyN for $\text{ESI}_{p95}$. **Implementation.** UNETR (Dice+CE); alignment $K$=4, $\tau$=0.07; fusion $\alpha$=2.0, $\beta$=1.0; deterministic seeds.

Table 3: Boundary contrast ($\text{ESI}_{p95}$; higher is better). Mean±sd over $n$=5.

| Method | Demons | SyN | VBH (ours) |
|---|---|---|---|
| $\text{ESI}_{p95}$ | $0.1183 \pm 0.0006$ | $0.1692 \pm 0.0004$ | $0.1918 \pm 0.0013$ |

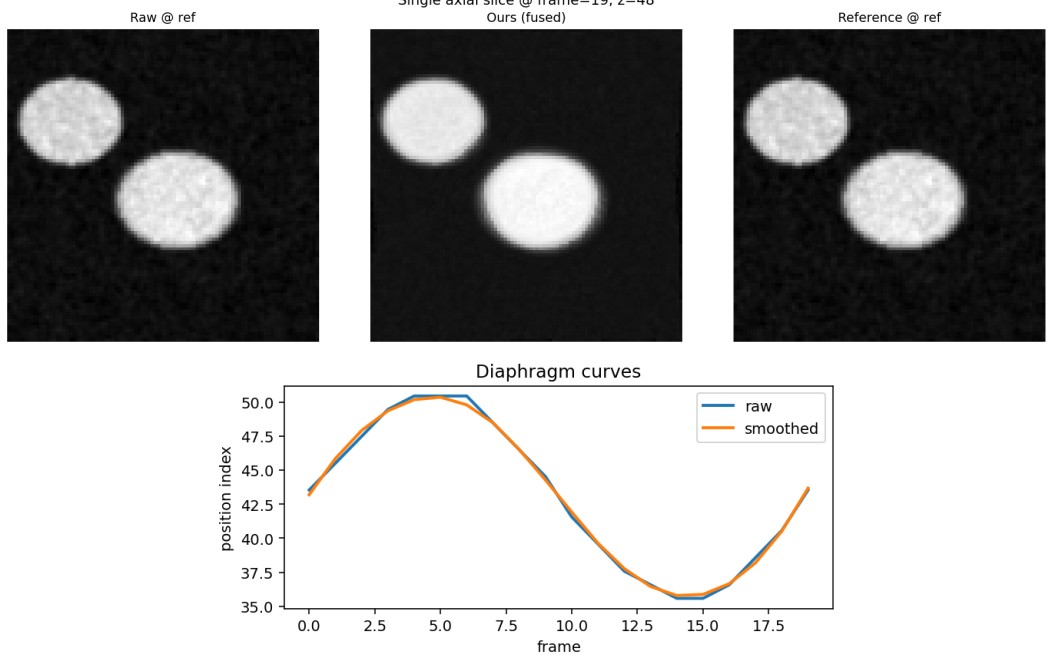

Figure 1: **Qualitative.** Top: Raw@ref, **VBH fused**, and Reference@ref (identical WL/WW and crop; 10 mm scale bar). VBH reduces diaphragm-induced blur/haloing. Bottom: surrogate $s(t)$ (raw vs EMA-smoothed).

## 4 Discussion, Limitations, Ethics & Reproducibility

VBH is a post-hoc, scanner-agnostic route to diaphragm-neutral volumes. Segmentation-guided fusion targets visible edges; alignment stabilizes performance with imperfect surrogates, aligning with multimodal rep learning and missing modalities. Limtations: small synthetic cohort ($n{=}5$), no test-time Jacobian checks. Reproducibility: macOS arm64 (MPS), PyTorch/MONAI FP32; alignment $K{=}4$, $\tau{=}0.07$; fusion $\alpha{=}2.0$, $\beta{=}1.0$; seeds per subject $1234{+}i$, config 42.

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
