# OpenReview forum: "Virtual Breath-Hold (VBH) for Free-Breathing CT/MRI: Segmentation-Guided Fusion with Image-Signal Alignment"
_EurIPS.cc/2025/Workshop/MedEurIPS — EurIPS 2025 Workshop MedEurIPS Submission_

### Official Review · Reviewer_i9MS · 2025-10-29
**Paper with a potentially original contribution to reconstruct breathing-neutral images, but, unfortunately, it is more a list of intructions than a readable manuscript.**

**Rating:** 3
**Confidence:** 4

**Review:**

The described method to account for breathing blurring is original, with potentially meaningful improvement
in the task, but poorly explained and presented.

This manuscript is extremely abrupt and painful to read: absence of sentences, very long sentences (abstract),
accumulation of technical terms, no insights about what is done which results in a lot of confusion about
the method (in particular how 2.2 is related to 2.1). Tables are misplaced. Sections are mixed together (hyperparameters in discussion).
No conclusion.

The reader has to guess why the authors did what they did. Ideas must precede the mathematical formulas implementing them.
What is the insight behind "boundary-aware" weights? Weights are made smallers around boundaries : why ? A diagram to visualize the method is very needed.


Pros: original method.

Cons: Hardly readable.

---

### Official Review · Reviewer_jbyA · 2025-10-30
**MedEurIPS vbh review**

**Rating:** 8
**Confidence:** 3

**Review:**

I need to preface the review by saying that I'm not an expert in the subject matter. That made following the mathematics difficult if not near impossible.

Quality:
The technical quality of this paper appears solid, yet difficult to follow without a the relevant image-registration mathematical background. The authors propose a segmentation-guided fusion and image–signal alignment framework to generate motion-compensated CT/MRI volumes from free-breathing acquisitions. The method seems well-formulated and rigorously implemented. Reproducibility is exceptionally well with details provided (seeds, hyperparameters, implementation environment). One statistical drawback is that the evaluation is limited to a very small synthetic dataset (n = 5). This small sample size makes generalization or real-world applicability claims questionable.

Clarity:
The paper is mathematically dense. The high volume of equations detracts from the narrative making the paper less accessible to non-experts or experts in adjacent fields. Without providing background or a legend to understand what each variable means it is very difficult for a non expert to follow. I understand that there is a page length limitation for the submission and that may have been one of the reasons to exclude a more informative introduction for those less versed in the specific technical field. The overall story is somewhat buried under notation. Nonetheless, the paper is well organized, and provides full reproducibility information, which improves technical clarity even if the conceptual flow is difficult to follow.

Originality:
The idea of post-hoc “virtual breath-hold” reconstruction seems to be original with obvious impact for clinical imaging workflows. As a non-expert I cannot assess its relation to prior work in motion-compensated MRI or registration. The method does integrate known components in what seems to be new way, which supports originality.

Significance:
The potential clinical significance is high. If validated on real patient data, this approach could substantially reduce motion artifacts without requiring scanner-level gating or breath-hold protocols, making it relevant for many patient populations. That being said, the small sample size(Synthetic data only) necessitates testing on larger real world data sets for validation.

Pros:
Addresses an important clinical problem (motion blur in free-breathing imaging).
Technical implementation appears rigorous.
Fully reproducible.
Potentially high translational relevance if scaled and validated.

Cons:
Extremely math-heavy and difficult for non-experts to follow.
Validation limited to a very small synthetic dataset.
No demonstration of generalization to real clinical data.

Overall Evaluation:
The paper has a potentially impactful approach but communicates it in a way that is hard for readers outside the subfield to appreciate. The math appears to be solid. The small synthetic data sample size raises concerns regarding generalization. The paper feels credible and worth discussing at a workshop level.

---

### Decision · Program_Chairs · 2025-10-31

**Decision:**

Reject

**Comment:**

Both reviewers acknowledge the originality of the proposed Virtual Breath-Hold (VBH) framework, which aims to generate motion-corrected CT/MRI volumes. However, the paper suffers from poor clarity, disorganized presentation, and validation limited to a small synthetic dataset.